# EXPLORING THE POTENTIAL OF LOW-BIT TRAINING OF CONVOLUTIONAL NEURAL NETWORKS

## ABSTRACT

In this paper, we propose a low-bit training framework for convolutional neural networks. Our framework focuses on reducing the energy and time consumption of convolution kernels, by quantizing all the convolutional operands (activation, weight, and error) to low bit-width. Specifically, we propose a multi-level scaling (MLS) tensor format, in which the element-wise bit-width can be largely reduced to simplify floating-point computations to nearly fixed-point. Then, we describe the dynamic quantization and the low-bit tensor convolution arithmetic to efficiently leverage the MLS tensor format. Experiments show that our framework achieves a superior trade-off between the accuracy and the bit-width than previous methods. When training ResNet-20 on CIFAR-10, all convolution operands can be quantized to 1-bit mantissa and 2-bit exponent, while retaining the same accuracy as the full-precision training. When training ResNet-18 on ImageNet, with 4-bit mantissa and 2-bit exponent, our framework can achieve an accuracy loss of less than $1\%$. Energy consumption analysis shows that our design can achieve over $6.8\times$ higher energy efficiency than training with floating-point arithmetic.

## 1 INTRODUCTION

Convolutional neural networks (CNNs) have achieved state-of-the-art performance in many computer vision tasks, such as image classification (Krizhevsky et al., 2012) and object detection (Redmon et al., 2016; Liu et al., 2016). However, deep CNNs are both computation and storage-intensive. The training process could consume up to hundreds of ExaFLOPs of computations and tens of GBytes of storage (Simonyan & Zisserman, 2014), thus posing a tremendous challenge for training in resource-constrained environments. At present, the most common training method is to use GPUs, but it consumes much energy. The power of a running GPU is about 250W, and it usually takes more than 10 GPU-days to train one CNN model on ImageNet (Deng et al., 2009). It makes AI applications expensive and not environment-friendly.

Table 1: The number of different operations in the training process (batch size $= 1$). Abbreviations: "EW-Add": element-wise addition, ; "F": forward pass; "B": backward pass.

| Op Name | Op Type | ResNet18 (ImageNet) | ResNet20 (CIFAR-10) |
|---------|---------|---------------------|---------------------|
| Conv (F) | Mul&Add | 2.72E+10 | 4.05E+07 |
| Conv (B) | Mul&Add | 5.44E+10 | 8.11E+07 |
| BN (F) | Mul&Add | 3.01E+07 | 1.88E+05 |
| BN (B) | Mul&Add | 3.01E+07 | 1.88E+05 |
| EW-Add (F) | Add | 1.49E+07 | 7.37E+04 |
| EW-Add (B) | Add | 1.20E+07 | 7.37E+04 |
| Weight Update (B) | Add | 1.12E+07 | 2.68E+05 |

Reducing the precision of NNs has drawn great attention since it can reduce both the storage and computational complexity. It is pointed out that the power consumption and circuit area of fixed-point multiplication and addition units are greatly reduced compared with floating-point ones (Horowitz, 2014). Many studies (Jacob et al., 2017a; Dong et al., 2019; Banner et al., 2018b) focus on amending the training process to acquire a reduced-precision model with higher inference efficiency.

Besides the studies on improving inference efficiency, there exist studies that accelerate the training process. Wang et al. (2018) and Sun et al. (2019) reduce the floating-point bit-width to 8 during the training process. Wu et al. (2018) implements a full-integer training procedure to reduce the cost but fails to get acceptable performance.

As shown in Tab. 1, Conv in the training process accounts for the majority of the operations. Therefore, this work aims at simplifying convolution to low-bit operations, while retaining a similar performance with the full-precision baseline. The contributions of this paper are:

1. This paper proposes a low-bit training framework to improve the energy efficiency of CNN training. We design a low-bit tensor format with multi-level scaling (MLS format), which can strike a better trade-off between the accuracy and bit-width, while taking the hardware efficiency into consideration. The multi-level scaling technique extracts the common exponent of tensor elements as much as possible to reduce the element-wise bit-width, thus improving the energy efficiency. To leverage the MLS format efficiently, we develop the corresponding dynamic quantization and the MLS tensor convolution arithmetic.

2. Extensive experiments demonstrate the effectiveness of our low-bit training framework. One only needs 1-bit mantissa and 2-bit exponent to train ResNet-20 on CIFAR-10 while retaining the same accuracy as the full-precision training. On ImageNet, using 4-bit mantissa and 2-bit exponent is enough for training ResNet-18, with a precision loss within $1\%$. Our method achieves higher energy efficiency using fewer bits than previous floating-point training methods and better accuracy than previous fixed-point training methods.

3. We estimate the hardware energy that implements the MLS convolution arithmetic. Using our MLS tensor format, the energy efficiency of convolution can be improved by over $6.8\times$, than the full-precision training, and over $1.2\times$ than previous low-bit training methods.

## 2 RELATED WORK

### 2.1 POST-TRAINING QUANTIZATION

Earlier quantization methods like (Han et al., 2015) focused on the post-training quantization, and quantized the pre-trained full-precision model using the codebook generated by clustering or other criteria (e.g., SQNR Lin et al. (2015), entropy Park et al. (2017)). Banner et al. (2018b) selected the quantization bit-width and clipping value for each channel through the analytical investigation. Jacob et al. (2017b) developed an integer arithmetic convolution for efficient inference, but it's hard to be used in training because the scale of the output tensor should be known before calculation. These quantization methods need pretrained models, and cannot accelerate the training process.

### 2.2 QUANTIZE-AWARE TRAINING

Quantize-aware training considered quantization effects in the training process. Some methods trained an ultra low-bit network like binary (Rastegari et al., 2016) or ternary (Li et al., 2016) networks, with a layer-wise scaling factor. Despite that the follow-up studies (Liu et al., 2020; Qin et al., 2019) have been proposing training techniques to improve the performance of binary networks, the extremely low bit-width still causes notable performance degradation. Other methods sought to retain the accuracy with relatively higher precision, such as 8-bit (Jacob et al., 2017a). Gysel et al. (2018) developed a GPU-based training framework to get dynamic fixed-point models. These methods focus on accelerating the inference process and the training process is still using floating-point operations.

### 2.3 LOW-BIT TRAINING

To accelerate the training process, studies have been focusing on design a better floating-point data format. Dillon et al. (2017) proposed a novel 16-bit floating-point format that is more suitable for CNN training, while Köster et al. (2017) proposed the Flexpoint that contains 16-bit mantissa and 5-bit tensor-shared exponent (scale), which is similar to the dynamic fixed-point format proposed by Gysel et al. (2018). Recently, 8-bit floating-point (Wang et al., 2018; Sun et al., 2019) was used with chunk-based accumulation and hybrid format to solve swamping.

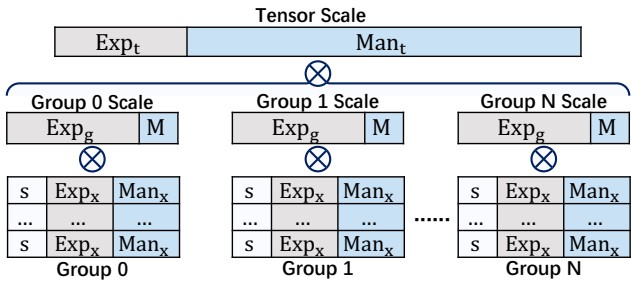

Figure 1: Illustration of the multi-level scaling (MLS) low-bit tensor data format.

Some studies used fixed-point in both the forward and backward processes (Zhou et al., 2016).Wu et al. (2018); Yang et al. (2020) implemented a full-integer training framework for integer-arithmetic machines. However, their methods caused notable accuracy degradation. Banner et al. (2018a) used 8-bit and 16-bit quantization based on integer arithmetic (Jacob et al., 2017b) to achieve a comparable accuracy with the full-precision baseline. But it's not very suitable for training as we discussed earlier. These methods reduced both the training and inference costs. In this paper, we seek to strike a better trade-off between accuracy and bit-width.

## 3 MULIT-LEVEL SCALING LOW-BIT TENSOR FORMAT

In this paper, we denote the filters and feature map of the convolution operation as weight (W) and activation (A), respectively. In the back-propagation, the gradients of feature map and weights are denoted as error (E) and gradient (G), respectively.

### 3.1 MAPPING FORMULA OF THE QUANTIZATION SCHEME

In quantized CNNs, floating-point values are quantized to use the fixed-point representation. In a commonly used scheme (Jacob et al., 2017b), the mapping function is $float = scale \times (Fix + Bias)$, in which $scale$ and $Bias$ are shared in one tensor. However, since data distribution changes over time during training, one cannot simplify the $Bias$ calculation as Jacob et al. (2017b) did. Thus, we adopt an unbiased quantization scheme,and extend the scaling factor to three levels for better representation ability. The mapping formula of our quantization scheme is

$$\boldsymbol{X}[i, j, k, l] = \boldsymbol{S_s}[i, j, k, l] \times S_t \times \boldsymbol{S_g}[i, j] \times \bar{\boldsymbol{X}}[i, j, k, l] \tag{1}$$

where $[\cdot]$ denotes the indexing operation, $\boldsymbol{S_s}$ is a sign tensor, $S_t$ is a tensor-wise scaling factor shared in one tensor, and $\boldsymbol{S_g}$ is a group-wise scaling factor shared in one group, which is a structured subset of the tensor. Our paper considers three grouping dimensions: 1) grouping by the 1-st dimension of tensor, 2) the 2-nd dimension of tensor, or 3) the 1-st and the 2-nd dimensions simultaneously. $S_t$, $\boldsymbol{S_g}$, and $\bar{\boldsymbol{X}}$ use the same format, which we refer to as $\langle E, M \rangle$, a customized floating-point format with E-bit exponent and M-bit mantissa (no sign bit). A value in the format $\langle E, M \rangle$ is

$$float = I2F(Man, Exp) = Frac \times 2^{-Exp} = \left(1 + \frac{Man}{2^M}\right) \times 2^{-Exp} \tag{2}$$

where $Man$ and $Exp$ are the M-bit mantissa and E-bit exponent, and $Frac \in [1, 2)$ is a fraction.

### 3.2 DETAILS ON THE SCALING FACTORS

The first level **tensor-wise scaling factor** $S_t$ is set to be an ordinary floating-point number ($\langle E_t, M_t \rangle = \langle 8, 23 \rangle$), which is the same as unquantized data in training to retain the precision as much as possible. Considering the actual hardware implementation cost, there are some restrictions on the second level **group-wise scaling factor** $S_g$. Since calculation results using different tensor groups need to be aggregated, using $S_g$ in an ordinary floating-point format will make more expensive conversions and operations necessary in the hardware implementation. We proposed two

hardware-friendly group-wise scaling scheme, whose formats can be denoted as $\langle E_g, 0\rangle$, and $\langle E_g, 1\rangle$. The scaling factor in the $\langle E_g, 0\rangle$ format is simply a power of two, which can be implemented easily as shifting on the hardware. From Eq. 2, a $S_g = I2F(Man_g, Exp_g)$ value in the $\langle E_g, 1\rangle$ format can be written as

$$S_g = \left(1 + \frac{Man_g}{2}\right) \times 2^{-Exp_g} = \begin{cases} 2^{-Exp_g} + 2^{-Exp_g-1} & Man_g = 1 \\ 2^{-Exp_g} & Man_g = 0 \end{cases} \tag{3}$$

which is a sum of two shifting, and can be implemented with small hardware overhead.

The third level scaling factor $S_x = I2F(0, Exp_x) = 2^{-Exp_x}$ is the **element-wise exponent** in $\bar{X} = S_x(1 + \frac{Man_x}{2})$, and we can see that the elements of $\bar{X}$ in Eq. 1 are in a $\langle E_x, M_x\rangle$ format. The specific values of $E_x$ and $M_x$ determine the type and the cost of the basic multiplication and accumulation (MAC) operation, which will be discussed later in Sec. 5.2.

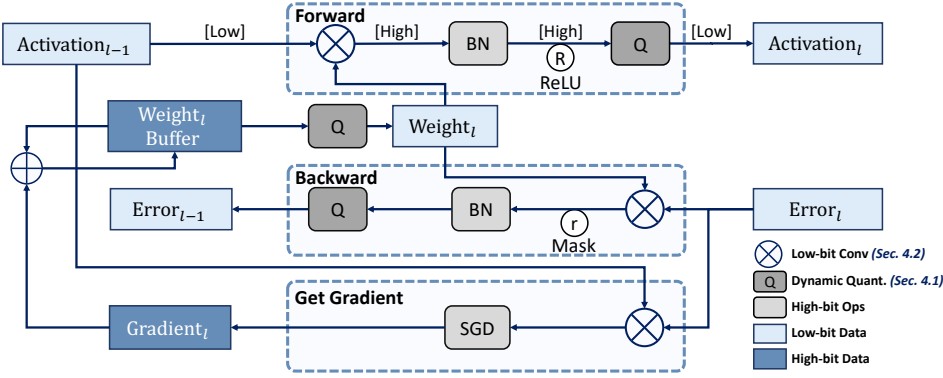

Figure 2: Computation flow of the proposed low-bit training.

## 4 LOW-BIT TRAINING FRAMEWORK OF CNN

As shown in Fig. 2, the convolution operation (Conv) is followed by batch normalization (BN), nonlinear operations (e.g. ReLU, pooling). Since Convs account for the majority computational cost, we apply quantization right before Convs in the training process, including three types: Conv(W, A), Conv(W, E), and Conv(A, E). Note that the output data of Convs is in floating-point format, and other operations operate on floating-point numbers. An iteration of this low-bit training process is summarized in Appendix Alg. 2, in which the major differences from the vanilla training process are the dynamic quantizaiton procedure $DynamicQuantization$ and the low-bit tensor convolution arithmetic $LowbitConv$.

### 4.1 DYNAMIC QUANTIZATION

The dynamic quantization converts floating-point tensors to MLS tensors by calculating the scaling factors $\boldsymbol{S_s}, S_t, \boldsymbol{S_g}$ and the elements $\bar{X}$, as shown in Alg. 1. $Exponent(\cdot)$ and $Fraction(\cdot)$ are to obtain the exponent (an integer) and fraction (a fraction $\in [0, 1)$) of a floating-point number. While calculating the quantized elements $\bar{X}$, we adopt the stochastic rounding (Gupta et al., 2015), and implement it by using a uniformly distributed random tensor $r \sim U[-\frac{1}{2}, \frac{1}{2}]$.

$$StochasticRound(x, r) = NearestRound(x + r) = \begin{cases} \lceil x \rceil & \text{with probability } x - \lfloor x \rfloor \\ \lfloor x \rfloor & \text{with probability } \lceil x \rceil - x \end{cases} \tag{4}$$

Note that Alg. 1 describes how we simulate the dynamic quantization process using floating-point arithmetic. While in the hardware design, the exponent and mantissa are obtained directly, and the clip/quantize operations are done by taking out some bits from a machine number.

---

**Algorithm 1:** Software-simulated dynamic quantization process

**Input:** $\boldsymbol{X}$: float 4-d tensor; $Axis$: grouping dimension; $\boldsymbol{R}$: $U[-\frac{1}{2}, \frac{1}{2}]$ distributed random tensor; $\langle E_g, M_g \rangle$: format of group-wise scaling factors; $\langle E_x, M_x \rangle$: format of each element

**Output:** $\boldsymbol{S_s}$: sign tensor; $S_t$: tensor group-wise scaling factors; $\boldsymbol{S_g}$: group-wise scaling factors; $\bar{\boldsymbol{X}}$: quantized elements

/\* **calculating scaling factors** \*/

1   $\boldsymbol{S_s} = Sign(\boldsymbol{X})$

2   $\boldsymbol{S_r} = Max(Abs(\boldsymbol{X}), axis = Axis)$           *// Group-wise maximum magnitude*

3   $S_t = Max(\boldsymbol{S_r})$           *// Tensor-wise maximum magnitude*

4   $\boldsymbol{S_{gf}} = \boldsymbol{S_r} \div S_t$           *// Group-wise scaling factors before quantization $< 1$*

5   $\boldsymbol{Exp_g}, \boldsymbol{Frac_g} = Exponent(\boldsymbol{S_{gf}}), Fraction(\boldsymbol{S_{gf}})$

6   $\boldsymbol{Exp_g} = Clip(\boldsymbol{Exp_g}, 1 - 2^{E_g}, 0)$           *// Clip $\boldsymbol{Exp_g}$ to $E_g$ bits*

7   $\boldsymbol{Frac_g} = Ceil(\boldsymbol{Frac_g} \times 2^{M_g}) \div 2^{M_g}$           *// Quantize $\boldsymbol{Frac_g}$ to $M_g$ bits*

8   $\boldsymbol{S_g} = \boldsymbol{Frac_g} \times 2^{\boldsymbol{Exp_g}}$           *// Group-wise scaling factors after quantization*

/\* **calculating elements** \*/

9   $\boldsymbol{X_f} = Abs(\boldsymbol{X}) \div \boldsymbol{S_g} \div S_t$           *// Dividing the scaling factors*

10   $\boldsymbol{Exp_x}, \boldsymbol{Frac_x} = Exponent(\boldsymbol{X_f}), Fraction(\boldsymbol{X_f})$

/\* **Quantize $\boldsymbol{Frac_x}$ to $M_x$ bits with underflow handling$^\dagger$** \*/

11   $E_{xmin} = 1 - 2^{E_x}$

12   $\boldsymbol{Frac_{xs}} = \boldsymbol{Frac_x} \times 2^{M_x}$ if not underflow, else $\boldsymbol{Frac_x} \times 2^{M_x - E_{xmin} + E_x}$

13   $\boldsymbol{Frac_{xint}} = Clip(StochasticRound(\boldsymbol{Frac_{xs}}, \boldsymbol{R})), 0, 2^{M_x} - 1)$

14   $\boldsymbol{Frac_x} = \boldsymbol{Frac_{xint}} \times 2^{-M_x}$ if not underflow, else $\boldsymbol{Frac_{xint}} \times 2^{-M_x + E_{xmin} - E_x}$

15   $\boldsymbol{Exp_x} = Clip(\boldsymbol{Exp_x}, E_{xmin}, -1)$

16   $\bar{\boldsymbol{X}} = \boldsymbol{Frac_x} \times 2^{\boldsymbol{Exp_x}}$           *// Elements after quantization*

17   **Return** $\boldsymbol{S_s}, S_t, \boldsymbol{S_g}, \bar{\boldsymbol{X}}$

---

$\dagger$: The underflow handling follows the IEEE 754 standard (Hough, 2019).

## 4.2   Low-bit Tensor Convolution Arithmetic

In this section, we describe how to do convolution with two low-bit MLS tensors. Denoting the input channel number as $C$ and the kernel size as $K$, the original formula of convolution in training is:

$$\boldsymbol{Z}[n, co, x, y] = \sum_{ci=0}^{C-1} \sum_{i=0}^{K-1} \sum_{j=0}^{K-1} \boldsymbol{W}[co, ci, i, j] \times \boldsymbol{A}[n, ci, x+i, y+j] \tag{5}$$

We take Conv(W, A) as the example to describe the low-bit convolution arithmetic, and the other two types of convolution can be implemented similarly. Using the MLS data format and denoting the corresponding values (scaling factors $\boldsymbol{S}$, exponents $\boldsymbol{Exp}$, fractions $\boldsymbol{Frac}$) of W and A by the superscript $^{(w)}$ and $^{(a)}$, one output element $\boldsymbol{Z}[n, co, x, y]$ of $Conv(W, A)$ can be calculated as:

$$\boldsymbol{Z}[n, co, x, y] = \sum_{ci=0}^{C-1} \sum_{i=0}^{K-1} \sum_{j=0}^{K-1} \left( S_t^{(w)} \boldsymbol{S_g^{(w)}}[co, ci] \bar{\boldsymbol{W}}[co, ci, i, j] \right) \left( S_t^{(a)} \boldsymbol{S_g^{(a)}}[n, ci] \bar{\boldsymbol{A}}[n, ci, x+i, y+j] \right)$$

$$= \left( S_t^{(w)} S_t^{(a)} \right) \sum_{ci=0}^{C-1} [\left( \boldsymbol{S_g^{(w)}}[co, ci] \boldsymbol{S_g^{(a)}}[n, ci] \right) \sum_{i=0}^{K-1} \sum_{j=0}^{K-1} \bar{\boldsymbol{W}}[co, ci, i, j] \bar{\boldsymbol{A}}[n, ci, x+i, y+j]]$$

$$= S_t^{(z)} \sum_{ci=0}^{C-1} \boldsymbol{S^{(p)}}[n, co, ci] \boldsymbol{P}[n, co, ci]$$

$$(6)$$

Eq. 6 shows that the accumulation consist of intra-group MACs that calculates $\boldsymbol{P}[n, co, ci]$ and inter-group MACs that calculates $\boldsymbol{Z}$. And the intra-group calculation of $\boldsymbol{P}[n, co, ci]$ is:

$$\boldsymbol{P}[n, co, ci] = \sum_{i,j=0}^{K-1} \left( \boldsymbol{Frac^{(w)}}[co, ci, i, j] \boldsymbol{Frac^{(a)}}[n, ci, i, j] \right) 2^{\left( \boldsymbol{Exp^{(w)}}[co, ci, i, j] + \boldsymbol{Exp^{(a)}}[n, ci, i, j] \right)}$$

$$(7)$$

Table 2: Comparison of low-bit training methods on CIFAR-10 and ImageNet. Single number in the bit-width stands for $M_x$, the corresponding $E_x$ is 0. "f" indicates that FP numbers are used.

| Dataset | Method | Bit-Width (W/A/E/Acc) | Model | Accuracy | Baseline |
|---|---|---|---|---|---|
| CIFAR-10 | (Banner et al., 2018a) | 1 1 2 - | ResNet-20 | 81.5% | 90.36% |
| | (Wu et al., 2018) | 2 8 8 32 | VGG-like | 93.2% | 94.1% |
| | (Rastegari et al., 2016) | 1 1 f32 f32 | ConvNet | 89.83% | 91.8% |
| | Ours | 4 4 4 16 | ResNet-20 | 92.32% | 92.45% |
| | | 2 2 2 16 | ResNet-20 | 90.39% | 92.45% |
| | | $\langle 1, 2 \rangle \; \langle 1, 2 \rangle \; \langle 1, 2 \rangle$ 16 | ResNet-20 | 91.48% | 92.45% |
| | | $\langle 2, 1 \rangle \; \langle 2, 1 \rangle \; \langle 2, 1 \rangle$ 16 | ResNet-20 | 91.97% | 92.45% |
| ImageNet | (Zhou et al., 2016) | 8 8 8 32 | AlexNet | 53.0% | 55.9% |
| | (Wu et al., 2018) | 2 8 8 32 | AlexNet | 48.4% | 56.0% |
| | (Yang et al., 2020) | 8 8 8 32 | ResNet-18 | 64.8% | 68.7% |
| | (Banner et al., 2018a) | 8 8 16 f32 | ResNet-18 | 66.4% | 67.0% |
| | (Sun et al., 2019) | $\langle 5, 3 \rangle \; \langle 5, 3 \rangle \; \langle 5, 3 \rangle$ f32 | ResNet-18 | 69.0% | 69.3% |
| | Ours | 8 8 8 32 | ResNet-18 | 68.5% | 69.1% |
| | | 6 6 6 16 | ResNet-18 | 67.6% | 69.1% |
| | | 4 4 4 16 | ResNet-18 | 66.5% | 69.1% |
| | | $\langle 1, 6 \rangle \; \langle 1, 6 \rangle \; \langle 1, 6 \rangle$ 16 | ResNet-18 | 68.0% | 69.1% |
| | | $\langle 2, 5 \rangle \; \langle 2, 5 \rangle \; \langle 2, 5 \rangle$ 32 | ResNet-18 | 68.3% | 69.1% |
| | | $\langle 2, 4 \rangle \; \langle 2, 4 \rangle \; \langle 2, 4 \rangle$ 32 | ResNet-18 | 68.2% | 69.1% |

where $\boldsymbol{Frac}$, $\boldsymbol{Exp}$ are fractions and exponents, whose precision is $(M_x + 1)$-bits and $E_x$-bits, respectively. Thus the intra-group calculation contains $(M_x + 1)$-bit multiplication, $(2^{E_x+1} - 4)$-bit shifting, and the $(2M_x + 2^{E_x+1} - 2)$-bit integer results are accumulated with enough bit-width to get the partial sum $\boldsymbol{P}$. And the accumulator has to be floating-point in some previous work (Wang et al., 2018; Sun et al., 2019), since they use $E_x = 5$. As for the inter-group calculation, each element in $\boldsymbol{S^{(p)}}$ is a $\langle E, 2 \rangle$ number obtained by multiplying two $\langle E, 1 \rangle$ numbers. Omitting the $n$ index for simplicity, the calculation can be written as:

$$\boldsymbol{Z}[co, x] = \sum_{ci=0}^{C-1} \boldsymbol{S^{(p)}}[co, ci] \boldsymbol{P}[x, ci] =$$

$$\sum_{ci=0}^{C-1} \begin{cases} \boldsymbol{P}[x, ci] 2^{-\boldsymbol{Exp^{(p)}}[co,ci]} & \text{if } \boldsymbol{Man^{(p)}}[co, ci] = 00 \\ \boldsymbol{P}[x, ci] 2^{-\boldsymbol{Exp^{(p)}}[co,ci]} + \boldsymbol{P}[x, ci] 2^{-\boldsymbol{Exp^{(p)}}[co,ci]-1} & \text{if } \boldsymbol{Man^{(p)}}[co, ci] = 01/10 \\ \boldsymbol{P}[x, ci] 2^{1-\boldsymbol{Exp^{(p)}}[co,ci]} + \boldsymbol{P}[x, ci] 2^{-\boldsymbol{Exp^{(p)}}[co,ci]-2} & \text{if } \boldsymbol{Man^{(p)}}[co, ci] = 11 \end{cases} \quad (8)$$

Due to the special format of $\boldsymbol{S^{(p)}}$, the calculation of the floating-point $\boldsymbol{Z}$ can be implemented efficiently on hardware, in which no floating-point multiplication is involved.

## 5 EXPERIMENTS

We train ResNet (He et al., 2016) on CIFAR-10 (Krizhevsky, 2010) and ImageNet (Deng et al., 2009) with our low-bit training framework. We experiment with the MLS tensor formats using different $\langle E_x, M_x \rangle$ configurations. And we adopt the same quantization bit-width for W, A, E, thus that hardware design is simple. The training results on CIFAR-10 and ImageNet are shown in Tab. 2. We can see that our method can achieve smaller accuracy degradation using lower bit-width. Previous study (Zhou et al., 2016) found that quantizing E to a low bit-width hurt the performance. However, our method can quantize E to $M_x = 1$ or 2 on CIFAR-10, with a small accuracy drop from 92.45% to 91.97%. On ImageNet, the accuracy degradation of our method is rather minor under 8-bit quantization (0.6% accuracy drop from 69.1% to 68.5%). In the cases with lower bit-width, our method achieves a higher accuracy (66.5%) with only 4-bit than Banner et al. (2018a) who uses 8-bit (66.4%). With $\langle 2, 4 \rangle$ data format, the accuracy loss is less than 1%. In this case, the bit-width

Table 3: Ablation study (ResNet-20 on CIFAR-10). "Div." means that the training failed to converge

| #group | $M_g$ | $E_x$ | $M_x = 4$ | $M_x = 3$ | $M_x = 2$ | $M_x = 1$ |
|--------|-------|-------|-----------|-----------|-----------|-----------|
| 1 | None | 0 | 90.02 | 85.68 | Div. | Div. |
| c | 0 | 0 | 91.54 | 88.35 | 82.29 | Div. |
| n | 0 | 0 | 91.78 | 89.62 | 80.71 | Div. |
| nc | 0 | 0 | 92.14 | 91.64 | 88.97 | 76.98 |
| nc | 1 | 0 | 92.37 | 91.73 | 90.39 | 82.61 |
| 1 | None | 0 | 90.02 | 85.68 | Div. | Div. |
| 1 | None | 1 | 91.67 | 90.11 | 84.72 | 70.4 |
| 1 | None | 2 | 92.32 | 92.34 | 91.58 | 90.32 |
| 1 | None | 3 | 92.66 | 92.41 | 92.47 | 92.04 |
| nc | 1 | 0 | 92.37 | 91.73 | 90.39 | 82.61 |
| nc | 1 | 1 | 92.52 | 92.16 | 91.48 | 89.97 |
| nc | 1 | 2 | 92.37 | 92.65 | 92.05 | 91.97 |

of the intermediate results is $2M_x + 2^{E_x+1} - 2 = 14$, which means that the accumulation can be conducted using 16-bit integers, instead of floating-points (Mellempudi et al., 2019).

## 5.1 ABLATION STUDIES

### 5.1.1 GROUPING DIMENSION

Group-wise scaling is beneficial because the data ranges vary across different groups. We compare the average relative quantization error of using the three grouping dimensions (Sec. 3.1) with $\langle 8, 1 \rangle$ group-wise scaling format and $\langle 0, 3 \rangle$ element format. The first row of Fig. 3 shows that the AREs are smaller when each tensor is split to $N \times C$ groups.

Furthermore, we compare these grouping dimensions in the training process. The results in the first section of Tab. 3 show that the reduction of AREs is important to the accuracy of low-bit training. And when tensors are split to $N \times C$ groups, the low-bit training accuracy is higher. And we can see that $M_g = 1$ is important for the performance, especially with low $M_x$ (e.g., when $M_x = 1$).

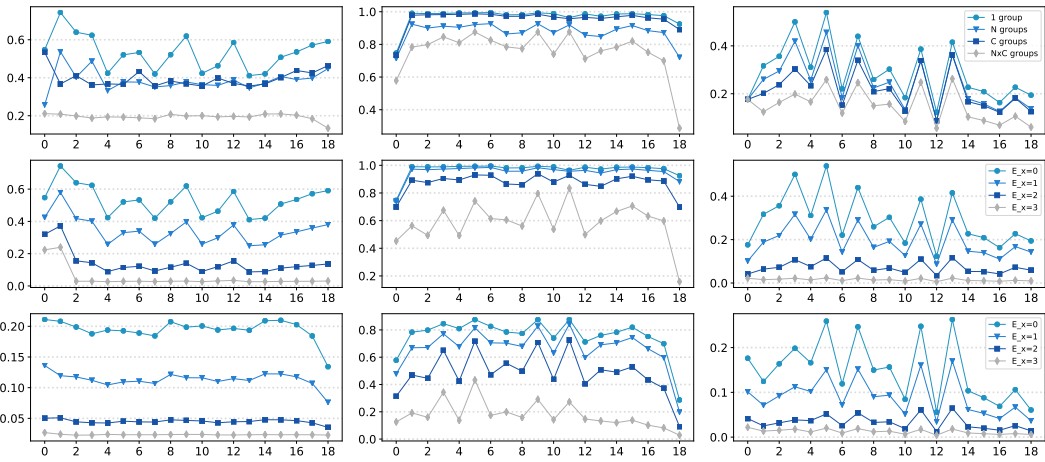

Figure 3: Average relative quantization errors (AREs) of W, E, A (left, middle, right) in each layer when training a ResNet-20 on CIFAR-10. X axis: Layer index. Row 1: Different grouping dimensions ($\langle 0, 3 \rangle$ formatted $\bar{X}$, $\langle 8, 1 \rangle$ formatted $S_g$); Row 2: Different $E_x$ ($\langle E_x, 3 \rangle$ formatted $\bar{X}$, no group-wise scaling); Row 3: Different $E_x$ ($\langle E_x, 3 \rangle$ formatted $\bar{X}$, $\langle 8, 1 \rangle$ formatted $S_g$, $N \times C$ groups).

### 5.1.2 ELEMENT-WISE EXPONENT

To demonstrate the effectiveness of the element-wise exponent, we compare the AREs of quantization with different $E_x$ without group-wise scaling, and the results are shown in the second row of Fig. 3. We can see that using more exponent bits results in larger dynamic ranges and smaller AREs. And with larger $E_x$, the AREs of different layers are closer. Besides the ARE evaluation, Tab. 3 also shows that larger $E_x$ achieves better accuracies, especially when $M_x$ is extremely small.

As shown in Fig. 3 Row 3 and Tab. 3, when jointly using the group-wise scaling and the element-wise exponent, the ARE and accuracy are further improved. And we can see that the group-wise scaling is essential for simplifying the floating-point accumulator to fixed-point, since one can use a small $E_x$ (e.g., 2) by using the group-wise scaling technique.

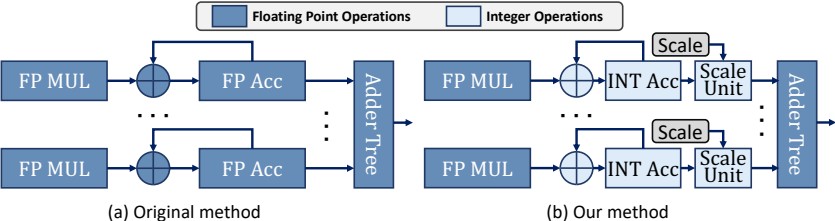

Figure 4: The convolution hardware architecture. (a) Previous studies (Mellempudi et al., 2019) developed low-bit floating-point multiplication (FP MUL) (e.g., 8-bit), but FP32 accumulations are still needed. (b) We not only makes FP MUL less than 8-bit, but also simplifies the local accumulator.

## 5.2 HARDWARE ENERGY ESTIMATION

Fig. 4 shows a typical convolution hardware architecture, which consists of three main components: local multiplication, local accumulation, and addition tree. Our algorithm mainly improves the local MAC. Compared with the full-precision design, we simplify the FP MUL to use a bit-width less than 8 and the local FP Acc to use 16-bit integer. According to the data reported by (Yang et al., 2020), the energy efficiencies are at least $7\times$ and $20\times$ higher than full-precision design, respectively.

While our method could significantly reduce the cost of the convolution, it also introduces some overhead: 1) Group-wise maximum statistics (Line 2) and scaling factors division (Line 9) in Alg. 1 accounts for the main overhead of **dynamic quantization**. The cost of these two operations is comparable with that of a batch normalization operation, which is relatively small compared with convolution since the number of operations is fewer by orders of magnitude (Tab. 1). 2) The **group-wise scaling factors** introduce additional scaling. Fortunately, when using the $\langle E_g, 0\rangle$ or $\langle E_g, 1\rangle$ format, we can implement it efficiently with shifting (see Eq. 3).

To summarize, the introduced overhead is small compared with the reduced cost. According to the numbers of different operations in the training process (Tab. 1) and the energy consumption of each operation (Appendix Tab. 3) (Horowitz, 2014), we can estimate that our convolution arithmetic is over $6.8\times$ more energy-efficient than full precision one when training ResNet (see Appendix for the details). Due to the simplified integer accumulator, our energy efficiency is at least 24% higher than other low-bit floating-point training algorithms (Mellempudi et al., 2019; Wang et al., 2018).

## 6 CONCLUSION

This paper proposes a low-bit training framework to enable training a CNN with lower bit-width convolution while retaining the accuracy. Specifically, we design a multi-level scaling (MLS) tensor format, and develop the corresponding quantization procedure and low-bit convolution arithmetic. Experimental results and the energy analysis demonstrate the effectiveness of our method.

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

## A  LOW-BIT TRAINING FRAMEWORK

---

**Algorithm 2:** The $t$-th iteration of low-bit training with vanilla SGD

---

**Input:** $L$: number of layers; $\boldsymbol{W}_t^{1:L}$: float weights; $\boldsymbol{A^0}$: inputs; $\boldsymbol{T}$: label; $lr$: learning rate
**Output:** $\boldsymbol{W}_{t+1}^{1:L}$: updated float weights
**/* forward propagation */**

1  **for** $l$ *in* $1 : L$ **do**

2  $\quad q\mathbf{W}^l = DynamicQuantization(\mathbf{W}^l)$

3  $\quad q\mathbf{A}^{l-1} = DynamicQuantization(\mathbf{A}^{l-1})$

4  $\quad \mathbf{Z}^l = LowbitConv(q\mathbf{W}^l, q\mathbf{A}^{l-1})$

5  $\quad \mathbf{Y}^l = BatchNorm(\mathbf{Z}^l)$

6  $\quad \mathbf{A}^l = Activation(\mathbf{Y}^l)$

7  $\frac{\partial loss}{\partial \mathbf{A}^l} = Criterion(\mathbf{A}^L, \boldsymbol{T})$

 

**/* backward propagation */**

**for** $l$ *in* $L : 1$ **do**

8  $\quad \frac{\partial loss}{\partial \mathbf{Y}^l} = \frac{\partial loss}{\partial \mathbf{A}^l} \times Activation'(\mathbf{Y}^l)$

9  $\quad \frac{\partial loss}{\partial \mathbf{Z}^l} = \frac{\partial loss}{\partial \mathbf{Y}^l} \times \frac{\partial \mathbf{Y}^l}{\partial \mathbf{Z}^l}$

10  $\quad q\mathbf{E}^l = DynamicQuantization(\frac{\partial loss}{\partial \mathbf{Z}^l})$

11  $\quad \mathbf{G}^l = LowbitConv(q\mathbf{E}^l, q\mathbf{A}^{l-1})$

12  $\quad \boldsymbol{W}_{t+1}^l = \boldsymbol{W}_t^l - lr \times \mathbf{G}^l$

13  $\quad$ **if** $l$ *is not* $1$ **then**

14  $\quad\quad \frac{\partial loss}{\partial q\mathbf{A}^{l-1}} = LowbitConv(q\mathbf{E}^l, q\mathbf{W}^l)$

15  $\quad\quad \frac{\partial loss}{\partial \mathbf{A}^{l-1}} = STE(\frac{\partial loss}{\partial q\mathbf{A}^{l-1}})$

16  **Return** $\boldsymbol{W}_{t+1}^{1:L}$

---

## B  EXPERIMENTAL SETUP

In all the experiments, the first and the last layer are left unquantized following previous studies (Zhou et al., 2016; Mellempudi et al., 2019; Sun et al., 2019). For both CIFAR-10 and ImageNet, SGD with momentum 0.9 and weight decay 5e-4 is used, and the initial learning rate is set to 0.1. We train the models for 90 epochs on ImageNet, and decay the learning rate by 10 every 30 epochs. On CIFAR-10, we train the models for 160 epochs and decay the learning rate by 10 at epoch 80 and 120.

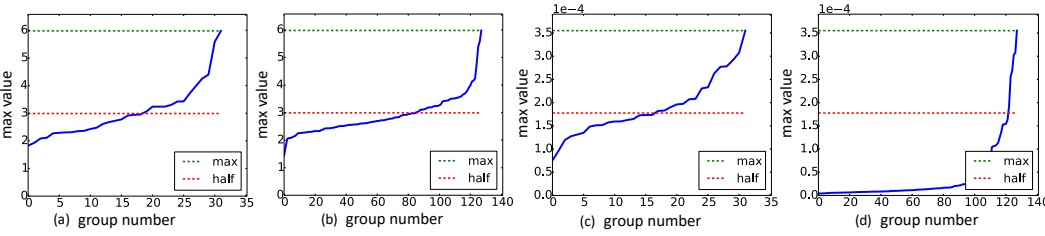

Figure 5: Maximum value of each group of A (left two) and E (right two). (a)(c): Grouped by channel; (b)(d): Grouped by sample.

## C    GROUP-WISE SCALING

Group-wise scaling is beneficial because the data ranges vary across different groups, as shown in Fig. 5. The blue line shows the max value in each group when A and E are grouped by channel or sample. If we use the overall maximum value (green lines in Fig. 5) as the tensor-wise scaling, many small elements will be swamped. And usually, there are over half of the groups, in which all elements are smaller than half of the overall maximum (red line). Thus, to fully exploit the bit-width, it is natural to use group-wise scaling factors.

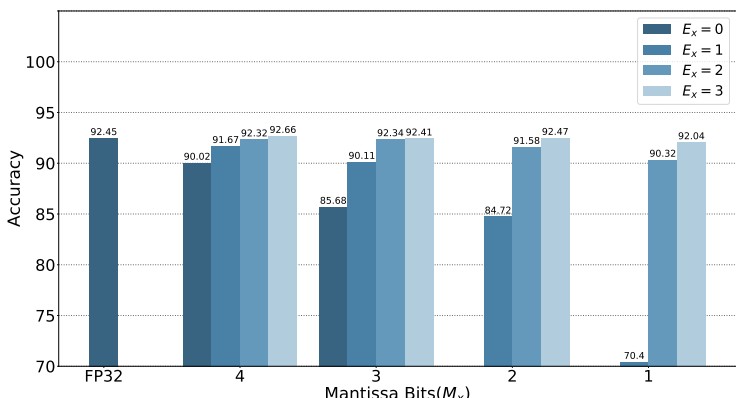

Figure 6: Performances with different $\langle E_x, M_x \rangle$ configurations, no group-wise scaling is used.

## D    ELEMENT-WISE EXPONENT

Fig. 6 shows the performances of training ResNet20 on CIFAR-10 with different $\langle E_x, M_x \rangle$ configurations. We can see that, when the mantissa bit-width $M_x$ is extremely low (e.g., 1), the element-wise exponent bit-width $E_x$ is essential for achieving an acceptable performance.

## E    ENERGY EFFICIENCY ESTIMATION

Tab. 4 (Horowitz, 2014) reported that the energy consumption of a FP32 multiplication (FP32 MUL) is about 4 times that of a FP32 addition (FP32 ADD). Denoting the energy consumption of FP32 ADD as $C$ and FP32 MUL as $4C$, according to (Yang et al., 2020), we can estimate the cost of FP8 MUL and INT16 ADD as $4/7C$ and $1/20C$, respectively. Then, using the operation statistics in Tab. 1, we can calculate the energy consumption of one training iteration, and estimate the energy efficiency improvement ratio:

$$EnergyRatio = \frac{4(\#MUL) + 1(\#LocalACC) + 1(\#TreeADD)}{4/7(\#MUL) + 1/20(\#LocalADD + \#Scale) + 1(\#TreeACC)} \approx 6.8$$

(9)

Table 4: The cost estimation of primitive operations with 45nm process and 0.9V (Horowitz, 2014).

| Params | Energy(pJ) | | Area($\mu m^2$) | |
|---|---|---|---|---|
| | Mul | Add | Mul | Add |
| 8-bit Fix | 0.2 | 0.03 | 282 | 36 |
| 16-bit Float | 1.1 | 0.40 | 1640 | 1360 |
| 32-bit Float | 3.7 | 0.90 | 7700 | 4184 |

In order to evaluate energy efficiency advantage more accurately, we have implemented the RTL design of the two MAC units in Fig.4, and used Design Compiler to simulate the area and power,

the power results are shown in Tab.5. We can see that the simulated results of RTL implemented is similar with our estimation above, both showing the energy efficiency of our framework.

Table 5: The power evaluation (mW) results of MAC units with different arithmetic with TSMC 65nm process and 100MHz clock, simulated by Design Compiler.

| Framework | Full Precision | Other Low-bit Training | Ours |
|---|---|---|---|
| MUL | 0.532 | 0.023 | 0.0192 |
| LocalACC | 0.140 | 0.140 | 0.0094 |

