# OpenReview forum: "Exploring  the Potential of Low-Bit Training of Convolutional Neural Networks"
_ICLR.cc/2021/Conference — Reject_

### Official Review · AnonReviewer3 · 2020-10-24
**Borderline Paper**

**Rating:** 6
**Confidence:** 4

**Review:**

This paper investigated the low-bit training problem and proposed a novel method, which can reduce the element-wise bit-width to simplify floating-point computations to nearly fixed-point. The major contribution can be summarized as follows: a multi-level scaling (MLS) tensor format, a dynamic quantization and the low-bit tensor convolution arithmetic. Experiments show the proposed method could strike a good trade-off between the accuracy and bit-width, while taking the hardware efficiency into consideration.

----Strengths:
- Experiment shows the proposed method can outperform the state-of-the-art methods
- There is abundant model analysis in this paper.
----Weaknesses:
- The paper is written in a very high-level manner. The paper could have benefitted with mode detailed explanations throughout, to aid comprehension.
- Many details of the method are unclear. For example: The definition of {I, j, k, l} in Formula 1 are unknown. Besides, in figure2, the exhibition of the proposed low-bit training computation flow is also confusing.
- On the algorithm side, The MLS format appears to be just a new combination of quantitative activation and multiple gain items. it is not clear how or why the proposed model should be more efficiency to reduce the element-wise bit-width compared with other methods? Is it just an observation/interpretation of the results?

---

> ### Author Response · Authors · 2020-11-23
> **Response to Reviewer 3**
>
> We sincerely appreciate your constructive and helpful comments. We initially address all your comments below:
>
> **A1:** For the confusion of Formula 1, we improved Section 3 to emphasize that Formula 1 is for a four-dimensional tensor with a shape of $l\times J\times K\times L$, using MLS format, and it is divided into $I\times J$ groups. Then, the concrete expressions of $S_t$, $S_g$ and $X$ are given more clearly. This part makes it clear how to express the original value by the quantization format. As for how to get the quantization format representation (i.e. how to calculate $S_t$, $S_g$ $Exp_x$, $Man_x$, etc.) from the original value of floating point, we give the calculation process in detail in Algorithm 1.
>
> **A2:** The rationality of MLS format can be explained by a simple theoretical analysis. Our method introduces multi-level scaling. Since the group-wise scaling factor extracts the common exponent of a group of data, it can reduce the bit-width of exponent of each quantized element as much as possible, which is very important for reducing the bit-width of convolution accumulator, as discussed in Section 4.2.

---

### Official Review · AnonReviewer1 · 2020-10-27

**Rating:** 4
**Confidence:** 4

**Review:**


Summary:
Efficient training is becoming a crucial research topic as deep learning models get deeper and more complex to improve model accuracy. The authors propose a low-bit floating point quantization method to reduce energy and time consumption during training. To enhance training efficiency, the authors suggest multi-level scaling (MLS) tensor format that enables low-bit training. MLS is designed to compute complex operations with lower-cost operators (e.g., shift, add) or lower-bit operators that can be hardware-friendly. This reviewer considers the proposed hardware-friendly MLS format as the main contribution of the manuscript.

This reviewer raises the following serious concerns.

- I have no idea whether this work can help efficient training in practice. Since there is no prior information on how many bits are required for training without serious accuracy degradation, any efficient training method needs to be robust to various learning settings and dataset configurations. This work, however, studies the optimal formats for particular models, such as ResNet models. It would be required to suggest a general data format and demonstrate that such a format can be applied to various DNN models.
- The authors argue that the proposed method achieves over 6.8x higher energy efficiency than training with floating-point (FP) arithmetic units. Unlike training with FP, the proposed method requires dynamic quantization for every batch. The authors need to present the computational overhead of the dynamic quantization with a sufficiently thorough analysis. If the cost is negligible, the ground should also be provided. Also, it would be better if the authors suggest the expected time reduction and space complexity compared to full-precision training. Even though Section 5.2 briefly addresses such overhead of dynamic quantization (as to be comparable with that of a batch normalization), this reviewer cannot estimate the overall benefit of the proposed method compared with the previous works.

Overall, this reviewer votes for rejection. An efficient training algorithm needs to be supported by detailed experimental results on various types of DNN models while any computational overhead should be reasonably addressed.

Minor comments:
- Section 2 needs to be focused on training while the comparisons with previous works can be elaborated in the experimental results.
- Table 1 is not relevant unless the authors want to suggest ResNet-specific accelerator designs.
- Similarly, Table 2 and 3, and Figure 3 are not necessary if the authors claim that the flow in Figure 2 can be general.

---

> ### Author Response · Authors · 2020-11-23
> **Response to Reviewer 1**
>
> Thank you for the insightful comments, which are helpful for us to further improve the quality of the paper! Please find the answers to your questions as below:
>
> **A1:** It is a difficult goal to choose the appropriate bit width for different models and datasets, to ensure the accuracy of the model and minimize the computational cost at the same time. On the one hand, the data distribution characteristics in different models on different datasets may be different, on the other hand, hardware design will also have to implement different bit width schemes, which greatly increases the design cost. For this problem, our goal is to make a trade-off and find a general format selection. Therefore, we add experiments of other network architectures on ImageNet. The results shows that <M, E> = <5, 2> is a general bit width scheme, which can be applied to various convolution neural networks, and the accuracy can be remained.
>
> **A2:** For the dynamic quantization overhead, we use the computation amount to evaluate. Taking the training of ResNet-18 on ImageNet as an example, the numbers of various operations in the usual floating-point training and low-bit training is shown in the table below, and the energy consumpution of each operation can be estimated as the values in the following table as well. Similar to the calculation method in Appendix E, the total energy consumption can be obtained by multiplying the amount of operations with the corresponding energy consumption. As can be seen in the results, because the number of operations of batchnorm and dynamic quantization are relatively small, the proportion of the total energy is also very small, which does not affect our low-bit training method to bring energy efficiency advantage in convolution operations. It is worth mentioning that in order to evaluate energy efficiency advantage more accurately, we have implemented the RTL design of the two MAC units in Figure 4 in the original manuscript, and used Design Compiler to simulate the area and power, the results are shown in Table 5 in the Appendix in the new version of manuscript.
>
> | Op Name    | Floating-Point |Training            |        || Low-Bit  |Training                   |        |
> | ---------- | ----------------------- | --------- | ------ || ---------------- | ----------------- | ------ |
> |            | Op Type                 | Op Amount | Energy | |Op Type          | Op Amount         | Energy |
> | Conv       | FloatMul                | 8.16E+10  | 302mJ  || IntMul           | 8.16E+10          | 43.2mJ |
> |            | FloatAdd                | 8.16E+10  | 73.4mJ || IntAdd           | 8.16E+10          | 3.67mJ |
> |            |                         |           |        || FloatAdd         | 9.07E+09          | 8.16mJ |
> | BN         | FloatMul                | 9.03E+07  | 334uJ  || FloatMul         | 9.03E+07          | 334uJ  |
> |            | FloatAdd                | 9.03E+07  | 81.3uJ || FloatAdd         | 9.03E+07          | 81.3uJ |
> | DQ         |                         |           |        || FloatMul         | 6.02E+7 + 2.24E+7 | 305uJ  |
> |            |                         |           |        || FloatAdd         | 6.02E+7 + 2.24E+7 | 74.4uJ |
> | EW-Add     | FloatAdd                | 2.69E+07  | 24.2uJ | |FloatAdd         | 2.69E+07          | 24.2uJ |
> |            |                         |           |        || FloatMul         | 2.69E+07          | 99.5uJ |
> | Weight-Add | FloatAdd                | 1.12E+07  | 10.1uJ || FloatAdd         | 1.12E+07          | 10.1uJ |
> | Sum        |                         |           | 376mJ  ||                  |                   | 56mJ   |
>
> **A3:** Our algorithm is not ResNet specific. Table 3 is an ablation study taking training ResNet-20 on CIFAR-10 as an example. Table 1 takes ResNet as an example to illustrate that convolution is the most important operation in CNNs. On the one hand, that's why we only focus on designing special data format and low-bit convolution arithmetic to reduce the multiplication and accumulation bit width in convolution. On the other hand, it also shows that dynamic quantization with small amount of operations will not greatly affect the overall energy efficiency improvement (as discussed in A2). The statistics of operands of other CNNs are basically consistent with the conclusion drawn from ResNet. For table 2, we add results of different networks and more existing low-bit training work, and we add the comparision of accumulation bit-width to emphasize the power consumption of accumulation. The results show that our scheme can achieve better energy efficiency and accuracy tradeoff than the previous work.

---

### Official Review · AnonReviewer2 · 2020-10-28
**The paper proposes a new tensor format referred to as multi-level scaling (MLS) for training deep neural networks in low precision. Overall, this was a difficult paper to read and understand. The results seem incremental.**

**Rating:** 4
**Confidence:** 4

**Review:**

The paper proposes a new tensor format referred to as multi-level scaling (MLS) for training deep neural networks in low precision. Dynamic quantization and MLS tensor arithmetic are used to enhance the effectiveness of MLS. Hardware energy efficiency gains are listed as 6.8X over full-precision and 1.2X over previous low-precision methods.

Comments and concerns are listed below:

1. This paper is a difficult read because of the disconnect between the figures used and the text. For example, the notation in Fig. 1 does not match the notation in the text describing Fig. 1 in 3.1 and 3.2. The symbol 'E' is used for both activation gradient and for the number of exponent bits. Typo ('quantizaiton') in line 7 of section 4. Algorithm 1 is too busy to be useful. Please abstract it out so the flow is clearer. The paper could do with a complete rewrite to tighten it up.

2. The main results in Table 2 shows that the improvement w.r.t. (Sun, 2019) is incremental. This seems consistent with the 1.2X energy efficiency gain mentioned elsewhere. Also, since this paper purports to contribute to low-bit training, why isn't there a comparison with the references (Dillon, Koester, Gysel, and Wang)? Instead Banner is included even though it falls in the category of post-training quantization.

3. It was not clear from Fig. 2 if the high-bit data and computations are in FP-32 or equivalent? Not sure what is meant.

Overall, this was a difficult paper to read and understand. The results seem incremental.

---

> ### Author Response · Authors · 2020-11-23
> **Response to Reviewer 2**
>
> We sincerely appreciate your constructive and helpful comments. We initially address all your comments below:
>
> **A1-1:** The schematic diagram of MLS format tensor in Figure1 does not correspond well with Formula 1. In order to make the relationship clearer, we will emphasize in Section 3 that Formula 1 is for a four-dimensional tensor with a shape of $I\times J\times K\times L$ and we use MLS format to divide it into $I\times J$ groups. Then we will change the "Tensor Scale" in Figure 1 to "$S_t$", "Group Scale" to "$S_g$" and change "Group N" to "Group [I, J]", "Group 0" to "Group [0,0]". And we add a mark of the size of one group: $K\times L$ values.
>
> **A1-2:** In this paper, we use "E" to denote the abbreviation of error, and the italicized "$E$" represents the bit-width of exponent, and "$Exp$" represents the value of exponent. In order to avoid confusion, we delete all the abbreviations E of error in the paper, and use Error instead (abbreviations of weight and activation are also deleted)
>
> **A1-3:** Algorithm 1 is indeed too complex, but it is necessary to show details. So we are going to add a flow chart to help to illustrate the algorithm more vividly.
>
> **A2:** We agree that our method should be compared with other low-bit training references (Dillon, Koester, Gysel, and Wang). Therefore, we will modify Table 1 and add these references to make a comparison. We can see that compared with other low-bit training work, our bit-width is reduced, especially the lower exponent can simplify the accumulator from floating point to fixed point, which has a great effect on reducing the power consumption of accumulation as discussed in Section 4.2. In order to illustrate our advantages, we implement the RTL design of the two MAC units shown in Figure 4. Through the comprehensive simulation of Design Compiler, we give a more detailed energy efficiency comparison, as shown in Table 4 in Appendix in the new version of manuscript.
>
> **A3:** The "high-bit data" in Figure 2 is in the format of FP32 in our current implementation, but in fact, our method is compatible with the work of using FP16 as the output of convolution and calculating BN (Sun), so it can also be replaced by FP16.

---

### Official Review · AnonReviewer4 · 2020-10-29
**Comparable results to state of art. Writing needs improvement.**

**Rating:** 5
**Confidence:** 3

**Review:**

This manuscript describes a new low-bit training framework as well as a new low-bit format and shows promising accuracy-precision trade-offs and better energy efficiency. The proposed method achieves no accuracy drop with 3-bit training on CIFAR dataset and 1% accuracy drop with 6-bit training on ImageNet dataset. Although the results are comparable to state-of-art works like "Hybrid 8-bit Floating Point", this paper has several drawbacks.

1. Section 3 provides detailed description of the MLS design. However, it is also very important to show a logic justification. Previous works will usually adopt metrics like mismatch probability across layers to show advantages of their design. Therefore, I would suggest authors to move section 5.1.1 (or Appendix C) to section 3 as a justification for MLS design.

2. Figure 2 looks like a typical quantization-aware training flow and I did not see any potential difference on the figure caused by MLS scheme. Therefore I wonder if it is right or necessary to put Figure 2 in this section.

3. It is not clear to me that how MLS format avoids 32-bit FP multiplication. As the first level scaling factor is still FP32, I suppose every activation still requires FP32 mul?

---

> ### Author Response · Authors · 2020-11-23
> **Response to Reviewer 4**
>
> Thank you for the insightful comments, which are helpful for us to further improve the paper! For your comments:
>
> **A1:** At present, the structure of the paper is to introduce the proposed technical points first, and then to prove its effectiveness through experiments. However, some insights of the proposed technical points in the method part can really help understanding, so we will add some ideas on why MLS format is proposed in Section 3.
>
> **A2:** Figure 2 is not a quantization-aware training flow. Quantization-aware training converts continuous floating-point numbers into discrete floating-point numbers, and then calculates them in floating-point format. More importantly, quantization-aware training only cares about weights and activations in forward path, so it uses floating-point format for back propagation. Our low-bit training framework really quantizes floating-point numbers into fixed-point numbers, and then uses the proposed low-bit convolution arithmetic to calculate. In addition, the convolution computation in the back propagation also uses the MLS format.
>
> **A3:** Our MLS Format is not aiming to avoid floating-point operation completely, but to use low-bit format for multiplication and accumulation in convolution, as discussed in Section 4.2. In the end, it is indeed necessary to multiply each value by tensor scale to get the real output. However, when the network layer are connected in sequence, the next layer needs quantization, too. Therefore, the real output value is not necessary and this multiplication operation is not required. This multiplication is only needed before element-wise addition.

---

### Decision · Program_Chairs · 2021-01-07
**Final Decision**

**Decision:**

Reject

**Comment:**

The authors propose a low-bit floating point quantization method to reduce energy and time consumption for deep learning training. Dynamic quantization and MLS tensor arithmetic are used to enhance the effectiveness of MLS. The motivation is clear and the efficient training is an important problem to address. However, the effectiveness of proposed method is not well justified and experimental results are less convincing.  In addition, the clarify of paper still needs to be further improved.